# *SmrtSwarm*: A Novel Swarming Model for Real-World Environments

Nikita Bhamu [1,†], Harshit Verma [1,†], Akanksha Dixit [2,*], Barbara Bollard [3] and Smruti R. Sarangi [1,*]

1   Department of Computer Science and Engineering, Indian Institute of Technology Delhi,
    New Delhi 110016, India; nikita.bhamu.cs518@cse.iitd.ac.in (N.B.); harshit.verma.mcs21@cse.iitd.ac.in (H.V.)
2   Department of Electrical Engineering, Indian Institute of Technology Delhi, New Delhi 110016, India
3   School of Earth, Atmospheric and Life Sciences, University of Wollongong, Wollongong 2522, Australia;
    bbollard@uow.edu.au
*   Correspondence: akanksha.dixit@ee.iitd.ac.in (A.D.); srsarangi@cse.iitd.ac.in (S.R.S.)
†   These authors contributed equally to this work.

**Abstract:** Drone swarms have gained a lot of popularity in recent times because, as a group, drones can perform highly intelligent tasks. Drone swarms are strongly inspired by the flocking behavior of birds, insects, and schools of fish, where all the members work in a coordinated manner to achieve a common goal. Since each drone is an independent entity, automating the control of a swarm is difficult. Previous works propose various swarming models with either centralized or distributed control. With distributed control, each drone makes its own decisions based on a small set of rules to accomplish swarm behavior, whereas in centralized control, one drone acts as the leader, who knows the final destination and the path to follow; it specifies the trajectories and velocities for the rest of the drones. Almost all the work in the area of swarming models follows Reynolds' model, which has three basic rules. For GPS-aided settings, state-of-the-art proposals are not mature enough to handle complex environments with obstacles where primarily local decisions are taken. We propose a new set of rules and a game-theoretic method to set the values of the hyperparameters to design robust swarming algorithms for such scenarios. Similarly, the area of realistic swarming in GPS-denied environments is very sparse, and no work simultaneously handles obstacles and ensures that the drones stay in a confined zone and move along with the swarm. Our proposed solution *SmrtSwarm* solves all of these problems. It is the first comprehensive model that enables swarming in all kinds of decentralized environments regardless of GPS signal availability and obstacles. We achieve this by using a stereo camera and a novel algorithm that quickly identifies drones in depth maps and infers their velocities and identities with reference to itself. We implement our algorithms on the Unity gaming engine and study them using exhaustive simulations. We simulate 15-node swarms and observe cohesive swarming behavior without seeing any collisions or drones drifting apart. We also implement our algorithms on a Beaglebone Black board and show that even in a GPS-denied setting, we can sustain a frame rate of 75 FPS, much more than what is required in practical settings.

**Keywords:** swarming models; Reynolds forces; stereo vision; depth map; decentralized control

## 1. Introduction

In the last couple of years, unmanned aerial vehicles (UAVs) have gained massive attention and are being used in diverse fields ranging from wildlife monitoring to aerial photography and agriculture [1–3]. The global commercial drone market alone was USD 19.89 billion in 2022 and is expected to grow at a CAGR (compound annual growth rate) of 13.9% from 2023 to 2030 [4]. Usually, these UAVs or drones are used in groups or swarms because a drone swarm tends to outperform single drones by leveraging their collective intelligence, increased versatility, and higher operational efficiency [5]. The concept of drone swarms is inspired by the flocking behavior of birds, animals, and insects, which exhibits a pattern and a similar general direction of motion for all its members. Broadly speaking, drone

swarming can be defined as the coordinated behavior of a group of autonomous drones that work together to achieve a common goal [6–8].

One of the earliest and most well-known flocking models was proposed by Reynolds [9]; it is also known as the boids model. It proposes three principles that govern the behavior of individual drones within a group. These rules are as follows: ❶ each drone in the swarm should maintain a minimum distance from its neighbors; ❷ it should align its velocity with the average velocity of its neighbors; ❸ it should move towards the center of mass of its neighbors. Reynolds demonstrated that these three rules controlling individual drone movement could result in complex collective behaviors such as flocking. The main aim of this model is to capture the self-organizing and coordinated motion observed in flocks of birds or schools of fish, where collective behavior emerges from localized interactions. As of today, it serves as the fundamental foundational algorithm that forms the basis for almost all drone swarming algorithms [6,8,10].

The Reynolds model is based on a distributed control algorithm, also known as self-organized flocking, where each UAV in the swarm decides its own movement. There are also models based on centralized control. In these models, the movement of UAVs is controlled either by an external agent or a specific drone within the swarm. The latter model is known as a leader–follower swarming model. The self-organizing swarm has the advantage of efficiency in terms of processing time since the work is divided among all the members, whereas the leader–follower structure is simpler and easier to implement and verify [11,12]. The leader drone guides the followers and offers additional control and coordination mechanisms. The follower drones can maintain a fixed distance and relative position with respect to the leader, ensuring that the swarm moves in a coordinated and synchronized manner. However, the problem with centralized control is the single point of failure. Hence, this paper proposes a hybrid model, *SmrtSwarm*, that combines the leader–follower and self-organized flocking models.

Braga et al. [6] suggest that for a leader–follower-based model, all the follower drones need to follow one more rule along with Reynolds' flocking rules, i.e., *Migration*, which forces the follower drones to migrate towards the leader drone. We integrate this behavior into *SmrtSwarm*. The drone swarms are designed for working in a real-world environment where the conditions may be adverse. For example, there may be obstacles, such as buildings, towers, etc., which may block a drone's path. The Reynolds model does not consider the presence of such obstacles. Hence, we propose including an additional obstacle avoidance rule that suggests alternative paths. However, this obstacle avoidance rule may lead to problems: the entire flock may disintegrate into smaller flocks with no interflock coordination, owing to obstacles. Olfati-Saber [13] identified this problem of fragmentation in flocking [9]. To avoid such a situation, the Olfati-Saber model proposes to define a boundary around the drones. Inspired by this method, we add a confinement rule in our model that forces all the drones to be confined to a predefined boundary.

To realize our model, each drone must be aware of its position and the position and movement of its neighbors. Therefore, the proposed model works only in a GPS-enabled environment where each flock member knows the position and velocity of others. But when operating in a real-world environment, such as in areas like mountain ranges, caves, congested urban areas, etc., access to a reliable GPS signal becomes a major hurdle [14–16]. In such scenarios, the swarm cannot rely on GPS for navigation. Much work has been carried out in building the swarming algorithms, but not a single framework has been provided that works on both the GPS-aided and GPS-denied environments; our proposed model has this capability.

This paper proposes a computer-vision-based strategy for achieving the flocking behavior in GPS-denied environments. We use a vision-based sensor to take pictures of the surrounding area and then analyze them to extract the required information. Previous works used ML-based models for segmenting and processing those images [17]. However, these methods require a significant amount of time and computing resources. Furthermore, many drones cannot afford to implement these algorithms for processing in every time

frame; as a result, we must use or create conventional algorithms instead. Therefore, we propose an image processing algorithm based on depth maps. No work has been presented that processes depth maps for computation of the movement of drones in the swarm. *SmrtSwarm* proposes a method to compute the depth maps of the images captured by drones and to find the neighboring drones and obstacles along with their distance from a reference drone. We also propose a novel algorithm to track the detected objects (drones and obstacles) over time. The swarming rules can be applied once all the necessary information is obtained. The code uses limited parallel processing to enhance its efficiency.

Our primary contributions in this paper are as follows:

1. We develop an enhanced Reynolds model that incorporates leader–follower behavior. The control is still distributed; however, the leader is a distinguished drone that knows the final destination.
2. We propose new Reynolds-like flocking rules that enable a swarm to navigate through GPS-aided environments containing physical obstacles while maintaining swarm behavior. The total processing time of our model is less than 1 ms on a popular embedded board.
3. We propose new flocking rules for GPS-denied environments as well. We develop a method to process depth maps quickly and process frames in around 13 ms ($\approx$75 fps) on a popular embedded board.

The paper is organized as follows. We discuss the background and related work in Section 2. Section 3 discusses the proposed swarming model. Section 4 shows the experimental results, and we finally conclude the paper in Section 5.

## 2. Background and Related Work

In this paper, we consider two kinds of scenarios. The first set of scenarios has an available GPS signal, which is arguably the most important input in a drone swarming system. The second set of scenarios does not rely on a GPS signal—they are more suitable for settings where GPS signals are weak, or places where jamming the GPS signal is a real possibility.

### 2.1. Swarming Models in an Environment with GPS Signals

Drone swarming is primarily inspired by the flocking behavior of birds. In general, flocking is a group behavior observed in birds, fish, and many other animals. It involves the coordinated movement of individuals within a group. To achieve this behavior, previous works propose various swarming models that enable drones in a swarm to communicate with one another and coordinate their movements [6,8–10]. These models specify certain rules for all the drones that guide them on how to react to the movement of other drones nearby. This way, each drone contributes to the overall group behavior. In general, the flocking process has five stages (see Figure 1). Each swarm member observes its surroundings and locates all other drones during the initial stage. Following that, it employs a neighbor selection strategy to select a set of neighbors who influence its movement. Every flocking model has a unique neighbor selection technique, such as choosing the $k$ closest drones as neighbors. The drone then detects other obstacles in its vicinity and tracks the obstacles as well as its chosen neighbors. Subsequently, the drone calculates the net force exerted on it by the selected neighbors and obstacles and adjusts its position accordingly. There exist various varieties of flocking behavior and resultant swarming models. The two broad categories of flocking behavior are *self-organized* and *leader–follower*.

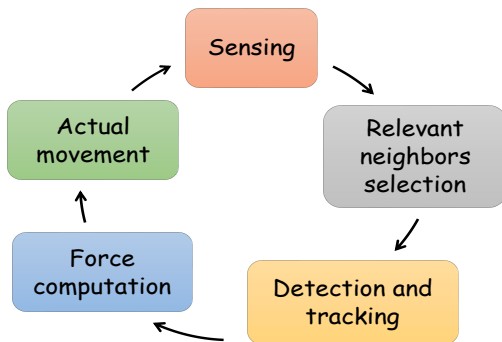

**Figure 1.** General workflow of a flocking model [6,18].

### 2.1.1. Self-Organized Swarming

Self-organized swarming, or decentralized flocking, is distinguished by the lack of *explicit leaders* within the group [19]. Instead, each group member follows simple principles to adjust its velocity based on its local interactions with other drones in the vicinity. Typically, these principles include maintaining a certain separation distance, aligning with the average direction of nearby drones, and moving toward the group's center of mass. Global flocking patterns arise from these local interactions. Fish schools and insect swarms exhibit this form of flocking behavior. Various papers have used different approaches to develop self-organized drone swarms [6,8–10]. Reynolds gave the firstever swarming model for self-organized flocking. Reynolds observed the natural flocking behavior and identified three simple rules that define the movement of each swarm member [9]. These rules are as follows:

(i)  **Cohesion:** Each swarm member must try to travel towards the group's center. This behavior is achieved by applying an attractive force between each flock member and the group's center of mass, which pulls the member towards the center (refer to Figure 2a).
(ii) **Separation:** Every member must keep a safe distance from its neighbors to prevent collisions. This is achieved by exerting a repulsive force between each flock member and its nearest neighbors (refer to Figure 2b).
(iii) **Alignment:** Every member in the swarm should try to match its neighbors' speed and direction. This behavior is achieved by exerting an attractive force between each flock member and its neighbors. This pushes the member's velocity closer to the group's average velocity (refer to Figure 2c).

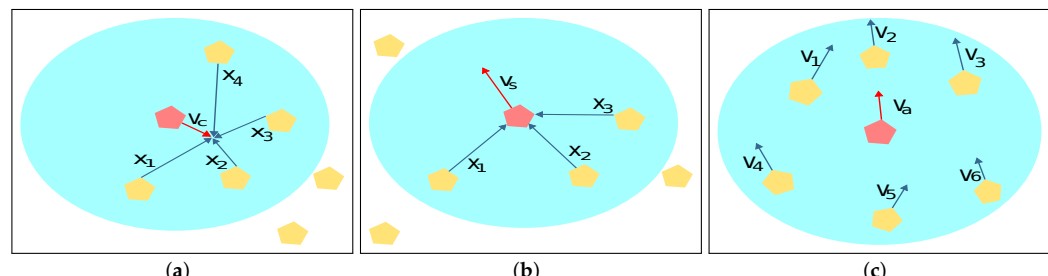

**Figure 2.** Reynolds' flocking principles: (**a**) cohesion, (**b**) separation, (**c**) alignment. $\vec{x}_i$ and $\vec{v}_i$ are the position and velocity of the $i^{th}$ drone, respectively. $\vec{v}_c$, $\vec{v}_s$, and $\vec{v}_a$ represent the cohesion, separation, and alignment velocity vectors, respectively. The final velocity of a drone is decided only by the drones inside the circular boundary.

If $\vec{v}_c$, $\vec{v}_s$, and $\vec{v}_a$ represent the velocity vectors resulting from the cohesion, separation, and alignment rules, respectively, then the final velocity $\vec{V}_f$ of a drone after incorporating

all three rules is given by Equation (1). Here, $r_c$, $r_s$, and $r_a$ are their respective weights in $\vec{V}_f$.

$$\vec{V}_f = r_c * \vec{v}_c + r_s * \vec{v}_s + r_a * \vec{v}_a \tag{1}$$

One point to note here is that the Reynolds model deals in velocities. It computes the velocity of each drone for the next frame based on the flocking rules and inherently exerts the required force to achieve that velocity. We adopt the same approach in the proposed model.

Eversham et al. [8] analyzed the classic Reynolds flocking model in detail and described the impact of the individual parameters on the observed flock behavior. Blomkvist et al. [10] proposed to use the Reynolds model to model flocking behavior in the case of prey escaping a predator attack. However, these models work in a very constrained environment with no obstacles. Braga et al. [6] considered the presence of obstacles and proposed an obstacle avoidance rule for swarm members. To detect obstacles, they used distance-based sensors. All these models rely on communication among drones, which may not be possible in real-world environments. Our proposed approach *SmrtSwarm* considers these real-world constraints and provides a robust solution that requires very little communication between drones. Communication is required only in a GPS-aided environment to broadcast its position, and no communication is required in a GPS-denied environment.

### 2.1.2. Leader–Follower Swarming

In leader–follower swarming, one or more group members undertake the role of a *leader*, while the remaining group members serve as followers [20]. The leaders determine the direction and pace of the flock, whereas the followers adjust their movements to maintain a certain distance or formation relative to the leaders. This flocking is commonly observed in avian colonies, where one or a few birds take the lead, and the remainder follow their movements. Bhowmick et al. [21] proposed a model with a leader–follower architecture with more than one leader; however, that number is fixed. They demonstrated how each member tends to move towards the center of the flock without colliding and still remains in the flock. However, it only operates in two dimensions and does not account for obstacles. Our proposed model *SmrtSwarm* **works in a 3D space, even with obstacles**.

Walker et al. [22] also proposed a leader–follower-based swarming model that considers multileader systems. However, the leaders are chosen dynamically during flight. Humans are needed to control all the leaders. If the swarm divides itself into clusters, each with a leader, and they become segregated, the operator must manually bring the leaders closer together each time this happens. This can occur frequently in an environment with obstacles. The *SmrtSwarm* model **addresses this issue by defining a confinement area around the leader and adding additional forces**. Unlike the other two models, Zheng et al. [23] proposed a flocking method with a single leader only. They also considered privacy concerns, such as hiding the leader if there is an adversary. However, their model does not define how to identify an adversary in a flock—its location or identity.

Reza Olfati-Saber [13] pointed out one disadvantage of the Reynolds flocking model: creating fragments in the swarm during flight time. Hence, the paper presented a method to make the swarm similar to an $\alpha$-lattice, ensuring no fragments are formed. However, the leader in their model is virtual and can change anytime during the flight. Hence, this required extra computation; also, it is not scalable to the environment when there is no GPS present. Our paper presents a simple yet effective algorithm that can be scalable to environments where GPS is an issue.

### 2.2. Swarming in a GPS-Denied Environment

As already discussed, most proposed models rely on GPS for location and velocity information. But relying solely on GPS for swarm navigation and coordination can pose challenges in real-world environments. GPS signals can be disrupted or lost due to various factors such as signal jamming, multipath interference, and natural obstructions such

as mountains, trees, and buildings. In such scenarios, swarms that heavily depend on GPS can face serious performance issues and may also suffer from complete failure [24]. To address this challenge, swarms must be designed to be more robust and resilient to GPS unavailability.

The drones thus need distance-sensing hardware [22,24]. Distance sensors are typically limited in range and accuracy. They also have reliability challenges while navigating in complex environments. In the natural world, birds and fishes rely on their sense of vision for perceiving distance [25]. A stereo camera is the most often used vision-based sensor for providing high-resolution images of the drone's surroundings and enabling it to recognize neighboring drones and obstacles. We are thus motivated to use such stereo cameras inspired by the natural world.

Previous works that deploy vision-based sensors in drone swarms use large computational neural networks to extract and use the information provided by the sensors [2,3,26]. In spite of this, state-of-the-art implementations mostly derive position information from such sensors. They seldom obtain good-quality velocity information that the Reynolds model requires. References such as [2,3] offer alternatives to the Reynolds model by employing a rigorous mathematically derived flocking algorithm that is based on the Laplace's equation—it relies on large convolutional neural networks (CNNs) for navigation in environments with obstacles.

For all the works that have been carried out in this area, most of them use machine learning methods to process images generated from a vision camera or use some other sensors to detect the distance and track other agents or obstacles, which is overhead for a drone for using multiple sensors or heavy computation [27]. And the papers which have used depth maps are meant for specific applications like flood level monitoring or processing depth maps with the Bayesian technique [28,29]. Our *SmrtSwarm* provides an efficient method of processing images without using any extra sensors or any machine learning methods to provide a distance of objects nearby to drones; in addition to detecting the objects, it also tracks the agent without requiring any extra hardware/software implementation. A brief comparison of related work is shown in Table 1. The summary of Section 2 is given in Figure 3.

❶ Both leader–follower and self-organizing swarming have their own benefits and drawbacks; we combine the best of both to create a hybrid swarming model that can work in environments with and without GPS signals.
❷ Existing works have one or more of the following limitations: they rely on GPS signals, they do not account for the presence of obstacles, they do not operate in three-dimensional space, they rely on communication between swarm members, and they use large CNNs that overwhelm the computational capacity of drones. *SmrtSwarm* **does not suffer from any of these limitations.**
❸ Vision-based sensors such as stereo cameras can be utilized for computing the positions and velocities of other drones in the vicinity. Using large CNNs for obtaining velocity or depth information from a 3D depth map of the environment is not a feasible idea. Drones have very limited onboard processing resources—there is thus a need to create bespoke depth map processing algorithms that are simple and fast. They should easily be able to run on popular embedded boards.

**Figure 3.** Insights from Section 2.

**Table 1.** A comparison of related work.

| Work | Year | Flock's Characteristics | | Environment | | Sensor Used | Algorithm |
| | | Leader–Follower | Self-Organized | GPS-Denied | Existence of Obstacles | | |
|---|---|---|---|---|---|---|---|
| Eversham et al. [8] | 2011 | × | ✓ | × | × | GPS | - |
| Blomqvist et al. [10] | 2012 | × | ✓ | × | ✓ | GPS | - |
| Barksten et al. [30] | 2013 | × | ✓ | × | × | GPS | - |
| Walker et al. [22] | 2014 | ✓ | × | ✓ | ✓ | Distance-based | - |
| Virágh et al. [31] | 2014 | × | ✓ | × | × | GPS | - |
| Bhowmick et al. [21] | 2016 | ✓ | × | × | × | GPS | - |
| Braga et al. [6] | 2016 | × | ✓ | × | × | GPS | - |
| Schilling et al. [26] | 2019 | × | ✓ | ✓ | × | Vision-based | ML-based |
| Zheng et al. [23] | 2020 | ✓ | ✓ | × | × | GPS | - |
| Schilling et al. [3] | 2021 | × | ✓ | ✓ | × | Vision-based | ML-based |
| Chen et al. [24] | 2022 | × | ✓ | ✓ | × | Distance-based | - |
| Schilling et al. [2] | 2022 | × | ✓ | ✓ | × | Vision-based | ML-based |
| *SmrtSwarm* | 2023 | ✓ | ✓ | ✓ | ✓ | Vision-based | Traditional CV |

## 3. Materials and Methods

### 3.1. SmrtSwarm in GPS-Aided Environments

Our swarming model is based on the conventional Reynolds model that incorporates the leader–follower behavior. In our implementation, each drone in the swarm considers all other drones to be its relevant neighbors even though this is not strictly necessary in larger settings. To improve the swarm's coordination and robustness, we suggest a few more novel swarming rules. The proposed rules are named ❶ migration, ❷ obstacle avoidance, and ❸ confinement. Note that when working in a GPS-aided setting, every swarm member is aware of its own position parameters, velocity, and tag, which are broadcasted to the other members. All swarming rules in the GPS-aided setting rely on this information.

Finally, the weighted sum of all the vectors generated by the newly introduced rules and the fundamental Reynolds rules are used to calculate the final velocity assigned to the drone.

The description, implementation, and mathematical representation of the rules are shown next. The mathematical representation is designed for a drone swarm having $n + 1$ drones, where one drone is the leader and the remaining $n$ drones are its followers. The mathematical notations are shown in Table 2.

**Table 2.** Glossary.

| Symbol | Meaning |
|---|---|
| $\vec{v}_i$, $\vec{x}_i$ | Velocity and position of the $i^{th}$ drone, respectively. |
| $\vec{x}_L$, $\vec{z}_j$ | Position of the leader drone and of the $j^{th}$ obstacle, respectively. |
| $\delta$ | Radius of the confined area around the leader. |
| $r_c$, $r_s$, $r_a$, $r_m$, $r_{ct}$, $r_{oa}$ | Weights of the cohesion, separation, alignment, migration, confinement, and obstacle avoidance rules, respectively, in the final velocity of the drone. |
| $\vec{v}_c$, $\vec{v}_s$, $\vec{v}_a$, $\vec{v}_m$, $\vec{v}_{ct}$, $\vec{v}_{oa}$ | Cohesion, separation, alignment, migration, confinement, and obstacle avoidance vector, respectively. |

### 3.1.1. New Rule: Migration Rule

The integration of the migration rule within *SmrtSwarm* enhances the functionality of our leader–follower-based model. In such models, the leader drone assumes complete trust from the follower drones, compelling them to faithfully adhere to its chosen path [1,6,9]. By introducing this novel rule, we address the challenges associated with coordinating a cohesive and goal-oriented flock.

The essence of the migration rule lies in its ability to facilitate the migration of follower drones towards the leader drone, thereby ensuring synchronized movement within the swarm (see Figure 4a). The fundamental objective is to eliminate deviations from the intended goal, as only the leader possesses the knowledge of the optimal route required to

reach the destination. This strategic alignment guarantees that each member of the flock remains focused and informed throughout the journey.

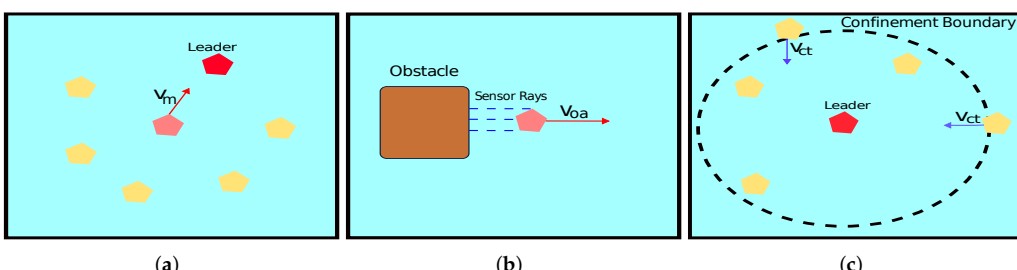

**Figure 4.** Proposed flocking rules in a GPS-aided environment: (**a**) migration; (**b**) obstacle avoidance; (**c**) confinement. Here, $\vec{v_m}$, $\vec{v_{ct}}$, and $\vec{v_{oa}}$ are migration, confinement, and obstacle avoidance vectors, respectively.

The migration vector is the directional vector from a given drone to the leader drone, which can be calculated by subtracting the position vectors of the respective drones. The rule is mathematically represented in Equation (2).

$$\vec{v_m} = \vec{x_L} - \vec{x_i} \tag{2}$$

### 3.1.2. New Rule: Obstacle Avoidance Rule

In real-world scenarios, the presence of obstacles, such as trees, buildings, poles, and other objects, poses an alarming challenge for drone swarms [32–34]. To tackle this challenge, we propose the inclusion of a new rule known as the obstacle avoidance rule in the *SmrtSwarm* model. Fundamental to our approach is the deployment of advanced distance sensors [35] on each drone within the swarm. These sensors can encompass either vision-based or infrared (IR) technology, providing the ability to detect obstacles within the environment. We carefully select and integrate these sensors, ensuring their suitability for obstacle detection tasks and their seamless integration with the overall swarm system.

We devise an obstacle avoidance rule that guides drones in creating new flight paths devoid of obstacles. Central to this rule is the fundamental directive for each member of the swarm to navigate in a direction away from the detected obstacle (refer to Figure 4b). However, we move beyond mere directional guidance. To enhance our obstacle avoidance strategy, we design the magnitude of the repulsive force in proportion to the inverse distance between the drone and the obstacle. This approach ensures that the repulsive force exerted along the line connecting the drone's center and the obstacle increases as the proximity to the obstacle decreases. This magnitude adjustment maximizes the likelihood of successfully steering the drones away from potential collisions and obstructions.

Our approach to obstacle detection and avoidance showcases our awareness of the real-world challenges faced by drone swarms. By integrating sensors and formulating an effective obstacle avoidance rule, we aim to ensure the safe navigation and successful completion of the swarm's mission even in the presence of obstacles.

Equation (3) shows the mathematical representation of this rule. It takes multiple obstacles into account.

$$\vec{v_{oa}} = -\sum_{j=1}^{n} (\vec{z_j} / |\vec{x_i} - \vec{z_j}|) \tag{3}$$

### 3.1.3. New Rule: Confinement Rule

The obstacle avoidance approach described in Section 3.1.2 may introduce potential issues where certain members of the swarm diverge from the rest of the group while avoiding obstacles. Furthermore, follower drones may surpass leader drones due to

prolonged exposure to these repulsive forces. To mitigate these concerns, we improve the suggested model by drawing ideas from the Olfati-Saber flocking model [13].

The Olfati-Saber model introduces the concept of a confinement area, which acts as a protective boundary surrounding the swarm, ensuring that no member, or drone, ventures outside of it. In our approach, we adopt a similar concept by defining a confined area around the leader drone. This confinement area serves as a virtual enclosure, preventing any subset of the flock from detaching or straying away from the main group (see Figure 4c).

According to the confinement rule, if any drone attempts to move outside the confinement area, a force is exerted to redirect it toward the leader drone. This redirection can be determined by subtracting the position vectors of the leader and the respective drone. The magnitude of this force is directly proportional to the extent to which the drone has deviated from the restricted area.

This confinement rule generates a nonzero vector only when a drone is outside the confinement zone, which is represented by a sphere with a radius of $\delta$ centered around the leader drone. When a member drone strays beyond this region, the confinement force acts to guide it back toward the leader, ensuring the cohesion and integrity of the flock.

By incorporating this confinement rule, we address the potential problem of swarm detachment and promote a collective behavior that preserves the cohesion and interdependence of the flock. The confinement force acts as a guiding mechanism, reinforcing the importance of staying within the predefined confinement area. This enhancement enhances the overall efficiency and coordination of the swarm, ensuring that no member drones deviate too far from the rest of the group.

Equation (4) provides the mathematical representation of the rule.

$$\vec{v_{ct}} = -((\vec{x_i} - \vec{x_L}) * max(0, |\vec{x_i} - \vec{x_L}| - \delta)) / |\vec{x_i} - \vec{x_L}| \tag{4}$$

Up till now, we have discussed the proposed flocking rules. Since *SmrtSwarm* combines these proposed rules with the basic Reynolds flocking rules and Section 2.1.1 only provides a brief description of the basic flocking rules, we provide their implementation details here.

### 3.1.4. Old Rule: Cohesion Rule

The cohesion vector (part of the original Reynolds model) tries to move the drone towards the swarm's centroid. Therefore, we need a vector pointing in that direction. We calculate this vector by averaging the neighboring drones' position vectors. Equation (5) provides the mathematical representation for this rule.

$$\vec{v_c} = \sum_{j=1}^{n} (\vec{x_j} / n) \tag{5}$$

### 3.1.5. Old Rule: Separation Rule

The separation vector tries to push the drone away from the neighboring drones; hence, a repulsive force needs to act between them along the line joining them. The direction of this force is from the neighbor towards the reference drone, and the magnitude of it is inversely proportional to the distance between the reference drone and the neighboring drone. The rule can be mathematically represented as Equation (6).

$$\vec{v_s} = \sum_{j=1}^{n} (\vec{x_i} - \vec{x_j}) / |\vec{x_i} - \vec{x_j}|^2 \tag{6}$$

### 3.1.6. Old Rule: Alignment Rule

The alignment vector tries to move the drone in the direction of the movement of the swarm. We can obtain the direction by taking the average of the velocities of all the drones

in the swarm and then moving the reference drone with that velocity. Equation (7) provides the mathematical representation for this rule.

$$\vec{v_a} = \sum_{j=1}^{n} (\vec{v_j}/n)$$

(7)

### 3.1.7. The Final Velocity

The final velocity $\vec{V_f}$ of a drone after incorporating all these rules is shown in Equation (8). Here, $r_m$, $r_{oa}$, and $r_{ct}$, $r_c$, $r_s$, and $r_a$ are the respective weights of these rules used in calculating $\vec{V_f}$. We are basically computing a linear weighted sum. The summary of Section 3.1 is given in Figure 5.

$$\vec{V_f} = r_c * \vec{v_c} + r_s * \vec{v_s} + r_a * \vec{v_a} + r_m * \vec{v_m} + r_{oa} * \vec{v_{oa}} + r_{ct} * \vec{v_{ct}}$$

(8)

> ❶ *SmrtSwarm* is a self-organizing model with a leader–follower behavior, increasing coordination and navigation within the drone swarm as opposed to the conventional Reynolds flocking model.
> ❷ We introduce a migration rule in the proposed flocking model, guiding follower drones to migrate toward the leader.
> ❸ A confinement rule is implemented, preventing subsets of the flock from detaching and maintaining overall cohesion.
> ❹ Obstacle avoidance is also addressed by equipping drones with sensors and implementing a rule that directs them away from detected obstacles.
> ❺ To obtain a balanced influence of various behavior, we use the weighted sum to integrate the effects of the proposed rules with the three fundamental Reynolds principles.

**Figure 5.** Insights from Section 3.1.

### 3.2. SmrtSwarm in GPS-Denied Environments

The model we propose in Section 3.1 requires the presence of a GPS while swarming, and also some communication regarding the drones' coordinates. The GPS signal helps each drone communicate its tags, velocity, and position with the other members of the swarm so that every drone can decide its motion accordingly. But GPS signal reception is not always possible in the real world, such as for indoor environments, dense urban areas, dense forests, security-sensitive environments, and places where there is a possibility of deliberate signal jamming [14–16]. Hence, we need to adopt *SmrtSwarm* for GPS-denied regions. For this, we use a computer-vision-based approach. We deploy a stereo camera on the drones to capture a specialized image of the environment that we shall refer to as the *depth map*. We need to then use a lightweight image processing approach to obtain the required information about the drones present in the field of view. We need to take into account that ML-based techniques are computationally expensive (refer to Section 2). Hence, we need to look at either ultrafast ML techniques or fast conventional algorithms. We were not able to find good candidate algorithms in the former class; hence, we opted for the latter class (i.e., conventional computer vision (CV) algorithms).

A *depth map* provides a pixelwise estimation of the depth or distance of objects from a particular viewpoint. It is typically represented as a 2D image, where each pixel corresponds to a depth value indicating the distance from the camera or the viewpoint. We use a bespoke algorithm on the produced depth map to obtain the neighboring drones' positions, velocities, and tags.

### 3.2.1. Object Detection in the Depth Map

We begin by computing the depth map of the current scene. One example of a depth map is shown in Figure 6. We make the following observations from the depth map:

❶　A depth map is a 2D matrix, where the value in each cell represents the depth of the relevant part of the object corresponding to it.

❷　The objects seen on the depth map form a cluster of pixels with similar pixel values. On the boundaries of these clusters, we can find a sudden change in pixel values.

❸　The values of the pixels belonging to objects far away from the reference point are very high.

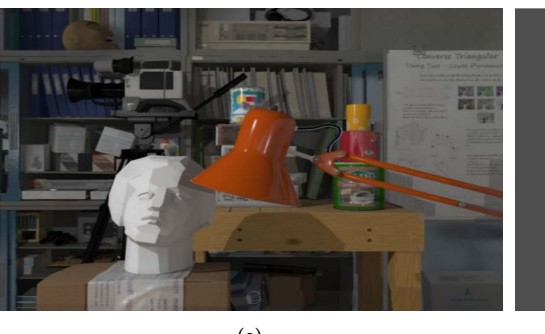 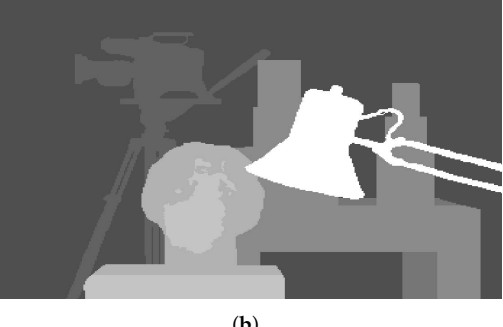

(**a**)　　　　　　　　　　　　　　　　　　　　　　　(**b**)

**Figure 6.** (**a**) Head and lamp image; (**b**) depth map of the image (adapted from the Tsukuba Stereo Vision dataset [36]).

We exploit these findings to detect the objects in the depth map. For the purpose of describing the object detection algorithm, we take one drone from the swarm as a reference drone. The suggested approach consists of two steps: (1) detecting objects in the depth map, and (2) determining if the identified objects are drones or obstacles. For the first step, we propose a depth-first search (DFS)-based approach. We observe that all objects that are in front of the reference drone form clusters in the 2D matrix corresponding to the depth map. To identify all the pixels that belong to a particular cluster (object), we traverse the 2D matrix (represented as a graph) using DFS. We ignore objects (clusters) that are far away from the reference point because they will not influence the movement of the drone. We also ignore very small clusters because they most likely do not correspond to drones. Both these behaviors are controlled by threshold parameters. The exact values of these two thresholds are given in Section 4.3.2. We start by identifying each cluster that is present in the depth map. These clusters might represent drones as well as obstacles; therefore, the next step is to determine whether they are drones or regular obstacles. They have different characteristics, as described in Section 4.3.2.

The identification of objects is not enough because the swarming model requires a few more details, such as the position and depth for applying the proposed rules. The drone's depth is determined by taking the pixel with the lowest value in the cluster that represents it. The component of the drone closest to the reference drone corresponds to the lowest pixel value. Additionally, to find its position, we compute the center of the drone by averaging the coordinates of all the pixels forming its cluster. For obstacles, we create a bounding box—a rectangular shape encompassing all the pixels in the cluster. This rectangle provides us with the obstacle's dimensions, and its depth is determined by the pixel with the lowest value within the cluster. By incorporating these data, we exert an obstacle avoidance force on the reference drone, ensuring that it steers clear of the surrounding obstacle.

### 3.2.2. Object Tracking

We need the velocities of the neighboring drones to calculate the alignment and confinement vectors; we need to know where a drone was in the previous and current frames. In the object tracking algorithm, there is only a single step: *tagging*. For tracking a drone, we associate it with a *unique tag*. Tagging also handles the problem of identifying objects that leave or newly enter the field of view (FoV) of a drone.

The tagging of the drones uses the insight that because the drones move slowly, the difference between their positions in successive frames will be less than a threshold (the exact value is mentioned in Section 4.3.2). The threshold depends on the cluster size representing the drone in the depth map. Since every drone in the swarm is of the same size, the neighbor drone closest to the reference drone has a larger cluster in the depth map and will move more than the others. Therefore, the threshold for movement in successive frames for this drone should reflect this fact. All of the drones' positions within the FoV are kept in lists. We maintain two such lists, one for the previous frame and the other for the current frame. While traversing these lists, the following cases may happen:

❶　A pair of positions in the list for the previous and current frames exists such that the difference between them is less than the threshold. Then we conclude that these are the positions of the same drone, and the drone is given the same tag in the current frame as it was in the previous frame.

❷　If a position in the list for the previous frame exists for which we are not able to find such a match (described in point 1) in the list for the current frame, then that position refers to a drone that recently left the FoV, and we do not issue a tag. In other words, if a tag found in the previous frame is not present in the current frame, then that drone has left the FoV of the reference drone.

❸　If a position in the list of the current frame exists for which we cannot find such a match in the list of the previous frame, then that position refers to a newly appeared drone in the FoV. It needs to be assigned a new tag.

The tags are initially assigned in ascending order. As more drones continue to enter the field of view, we increment a counter and assign the new value as the new drone's tag. In this way, we track the drones. We assume that there are no moving objects in the environments except for swarm members. In other words, this paper only considers static obstacles, which do not need to be tracked. After the completion of the steps mentioned in Sections 3.2.1 and 3.2.2 for object detection and tracking, we gather all the required information about the neighborhood and obstacles in the environment.

We still need a notional leader drone here. It is the drone that is at the front of the swarm and cannot see any drones in its front-facing camera (towards the direction of motion). It basically knows where to go. It either has a GPS or, using visual guidance, it knows the path. The rest either *implicitly* follow it or have their own guidance system. This means that the leader drone has special hardware that allows it to set the course; the rest follow the leader. For instance, if the drones are tracking wildlife and they can see a pack of deer, then they can all decide (independent of each other) to follow the pack and not the leader. All the swarming rules are still required to ensure that they behave as a swarm. It turns out that we need to make some alterations to the Reynolds rules and also propose a new rule for this setting.

### 3.2.3. Flocking Rules in GPS-Denied Environments

All the flocking rules proposed for a GPS-aided environment in Section 3.1 are applicable to this case except one—the migration rule. Since all drones have identical physical characteristics and, as a result, have the same kind of depth map projection, it is impossible to distinguish between a leader and a follower by looking at the depth map. This is why we skip the migration rule, which makes follower drones move towards the leader.

All the rules, which use only the position information of the drones forming the swarm and the obstacles in the environment, are implemented in the same way as mentioned for GPS-aided environments in Section 3.1 (refer to Figure 7). The rules falling into this category are the cohesion, separation, and obstacle avoidance rules. The alignment and confinement rules use velocity and tags, respectively. The algorithmic implementation of these rules in a GPS-denied environment needs to slightly change. This is because finding the velocities and tags is more complex than deducing the positions.

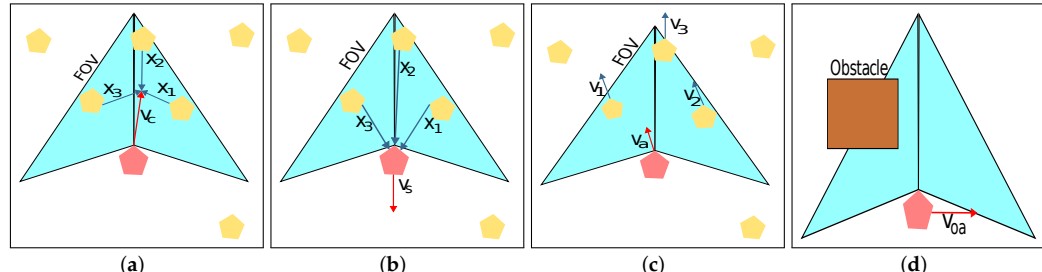

**Figure 7.** Proposed flocking rules in a GPS-denied environment: (**a**) cohesion; (**b**) separation; (**c**) alignment; (**d**) obstacle avoidance. Here, FoV represents the field of view of the reference drone. $\vec{x}_i$ and $\vec{v}_i$ are the position and velocity of the $i^{th}$ drone. $\vec{v}_c$, $\vec{v}_s$, $\vec{v}_a$, and $\vec{v}_{oa}$ represent cohesion, separation, alignment, and obstacle avoidance vectors.

Alignment Rule for GPS-Denied Environments

According to this rule, a drone needs to move in the direction given by the average velocity of drones present in its field of view. We store the position vectors of all the drones in the FoV for the previous and current frames in two lists. We store the tags assigned to drones for both frames, too. To find the velocity of a drone, we subtract the position vectors of the current and previous frames. We can find the velocity of only that drone that is present in both the current and previous frames. The drones which newly appeared in the FoV or recently left the FoV will not contribute to this. We then need to move the reference drone in the same direction as the mean average velocity (note: it is a vector). The complete flow is shown in Algorithm 1.

---

**Algorithm 1:** Alignment.

---

1 **Function** `Alignment()`:
2     $\vec{v}_a \leftarrow 0$
3     $counter \leftarrow 0$
4     $valNeighbours \leftarrow 0$         `/* Initialize the count of valid neighbors */`
5     **while** $counter < currPositions.size$ **do**
        `/* Get the current position and tag of the drone */`
6         $currPos \leftarrow currPositions[counter]$
7         $currTag \leftarrow currTags[counter]$
8         $i \leftarrow 0$         `/* Initialize the inner loop counter */`
9         **while** $i < prevTags.size$ **do**
10             **if** $currTag == prevTags[i]$ **then**
11                 $prevPos \leftarrow prevPositions[i]$ `/* Get the previous position of the corresponding drone */`
12                 $\vec{v}_a = \vec{v}_a + (currPos - prevPos)$ `/* Add the difference of positions, i.e., their velocity in a unit time frame to the alignment vector */`
13                 $valNeighbours = valNeighbours + 1$
14                 **break**
15             **end**
16             $i = i + 1$         `/* Move to the next tag in prevTags list */`
17         **end**
18         $counter \leftarrow counter + 1$ `/* Move to the next drone */`
19     **end**
20     $\vec{v}_a \leftarrow \vec{v}_a / valNeighbours$ `/* Normalize the alignment vector */`
21     **return** $\vec{v}_a$

---

Confinement Rule for GPS-Denied Environments

In a GPS-denied setting, a drone is said to be out of the confined area if no other drones are within its field of view. This can easily be found because we maintain a list of position vectors of all such drones (refer to Section 3.2.3). If a drone is outside the confined area, we assign it a velocity in the opposite direction of its current velocity $v_t$, known as the confinement velocity. The drone will continue in this direction until it detects a neighbor within its field of view. We cannot, however, just let the drone continue because it may not find any drones, even on this route. To prevent this, we limit the number of frames ($\kappa$) for which the drone can move in the opposite direction. If, within this limit, the drone does not encounter any drone within its field of view, it returns to its previous direction, i.e., the direction of $v_t$, and moves $2\kappa$ steps, then it moves $4\kappa$ steps in the opposite direction, and so on and so forth, until it sees other drones. Algorithm 2 shows the complete implementation. In our exhaustive simulations, we never had a case where a drone became lost, even in an environment with obstacles.

---

**Algorithm 2:** Confinement.

---

1 **Function** `Confinement()`:
2     $\vec{v_{ct}} \leftarrow 0$
    `// If there are neighboring drones in the FoV`
3     **if** *currPositions.size* > 0 **then**
4         *confinementCounter* = 0
5         *limit* = $\kappa$
6     **end**
7     **else**
        `/* Set the confinement vector opposite to the previous velocity */`
8         $\vec{v_{ct}} \leftarrow (prevVelocity.x, prevVelocity.y, -1 * prevVelocity.z)$
9         *confinementCounter*+ = 1
        `/* If the confinement counter exceeds the limit */`
10         **if** *confinementCounter* > *limit* **then**
11             *limit* = *limit* × 2                     `/* update the limit */`
12         **end**
13     **end**
14     **return** $\vec{v_{ct}}$

---

### 3.3. Workflow of the Proposed Model

Figure 8 shows the complete workflow of *SmrtSwarm* in a GPS-denied environment. For each frame, we compute a depth map, detect all the objects within it, and then compute their relative positions. We track the drones using information from the previous frame, and then compute the velocity of all the drones, and their tags. This information is used to compute all the velocities (yielded by the different rules), and the final target velocity is a weighted sum of all the individual velocities (similar to Equation (8)). The summary of Section 3.2 is given in Figure 9.

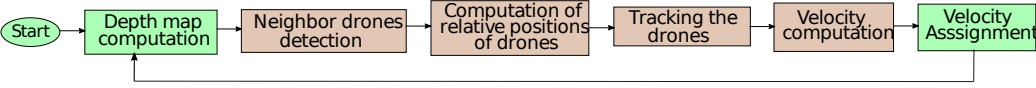

**Figure 8.** Workflow of *SmrtSwarm* in GPS-denied environments.

❶ In this model, we address the limitations of GPS signal reception in real-world environments and propose a computer-vision-based approach using cameras and depth maps to overcome this limitation.
❷ The migration rule from the GPS-aided model needs to be excluded in a GPS-denied environment due to its inability to distinguish between leaders and followers based on depth maps.
❸ Drones can become lost in a GPS-denied environment. Thus, the confinement rule needs to also have an element of searching that will allow a drone to rejoin the swarm if it temporarily moves out.

**Figure 9.** Insights from Section 3.2.

## 4. Results and Analysis

### 4.1. Simulation Setup

We implement *SmrtSwarm* on Unity, a popular cross-platform game development engine. It has a lot of features and prebuilt elements for creating custom environments. We added our code in C# for simulating a drone swarm [37,38] to it. We also experimented with the Unreal engine [39] but found it to be far slower than Unity, especially when the number of drones in the flock is increased. Other than visual effects, it was not adding any additional value. Hence, we opted for Unity version 2020.3.40f1 for simulating our system (similar to [37]). We use *C#* version 11.0 [40] for implementing the algorithms. A few simulation environments were created using Unity assets, and a few were purchased from the Unity store, which contains urban settings with both low- and high-rise towers and buildings [41,42]. The configuration details of the simulator are shown in Table 3. The simulated scenes and the drones placement are shown in Figure 10. In the literature on drones, using simulators for studying the behavior of large drone swarms is the standard practice [2,3]. Given that we do not have any other direct competitor that implements swarming with obstacle avoidance in GPS-aided and GPS-denied environments (see Table 1), we did not perceive the need to implement any state-of-the-art algorithm and compare the results with our paper.

**Table 3.** Platform configuration.

| Parameter | Value |
| --- | --- |
| Simulator | Unity 2020.3.40f1 |
| Operating system | Windows 10 |
| Main memory | 1 TB |
| RAM | 32 GB |
| CPU | Intel(R) Core(TM) i7-8700 CPU @ 3.20 GHz |
| GPU | NVIDIA Ge-Force GT 710 |
| Video memory | 2 GB |

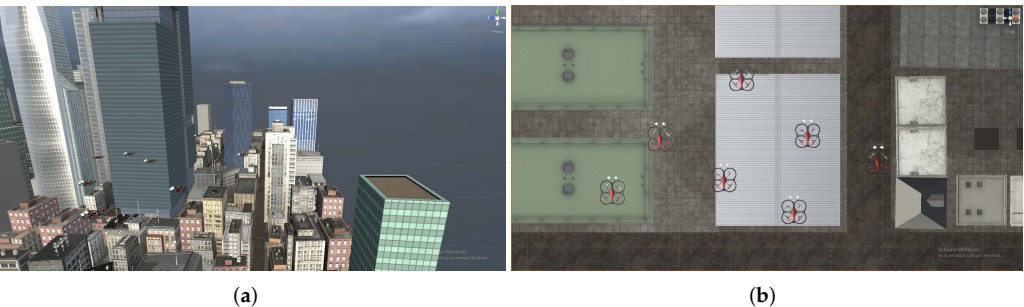

(**a**)                                    (**b**)

**Figure 10.** (**a**) Simulation scene; (**b**) drone placement.

*4.2. Setting the Hyperparameters (Coefficients in the Equations)*

Recall that in Equation (8) in Section 3, we assigned weights to each component velocity vector for computing the final velocity vector. In this section, we shall evaluate the impact of these weights on drone swarming and find their best possible values.

To find the optimum value of each hyperparameter, we first assigned equal weights to each hyperparameter and then observed which force is acting more aggressively and which is not. We then fixed all hyperparameters except one and tried to discover its optimum value (creating a Nash equilibrium). For instance, when determining $r_s$, we tried to determine how quickly drones are moving apart from one another. The optimum value was achieved when they moved at such a speed that they did not collide, yet still remained in the swarm. We then set this hyperparameter around that value and modified other hyperparameters one by one. We are basically computing a Nash equilibrium here, where the parameters are the players and the performance is the utility function.

We ran this experiment several times and tried many different parameter perturbations. For each experiment, we assessed its *performance*, which is defined as follows. It is a tuple comprising an integer (number of collisions or #collisions) and a Boolean value (whether any drone escaped the confinement zone).

Tables 4 and 5 show the obtained results for a scene with an enabled and disabled GPS signal, respectively. Note that in our simulations we did not observe any collisions because the hyperparameters were chosen correctly. Specifically, we make the following observations from the results:

**Table 4.** Effect of the weights on the overall performance for a GPS-aided environment.

| Experiment No. | Weights | | | | | | Performance | |
|---|---|---|---|---|---|---|---|---|
| | $r_c$ | $r_s$ | $r_a$ | $r_m$ | $r_{ct}$ | $r_{oa}$ | #Collisions | Confined |
| 1 | 10 | 1.0 | 1.0 | 1.0 | 2 | 1.0 | 3 | ✕ |
| 2 | 60 | 1.1 | 1.5 | 1.0 | 15 | 5.0 | 5 | ✓ |
| 3 | 75 | 1.2 | 1.0 | 1.1 | 25 | 5.0 | 3 | ✓ |
| 4 | 77 | 1.2 | 1.0 | 1.1 | 21 | 5.0 | 1 | ✕ |
| **5** | **77** | **1.2** | **1.0** | **1.1** | **21** | **5.0** | **0** | **✓** |
| 6 | 78 | 1.0 | 1.2 | 1.0 | 24 | 4.9 | 1 | ✓ |
| 7 | 80 | 1.1 | 1.0 | 1.2 | 23 | 5.1 | 2 | ✓ |
| 8 | 80 | 11.0 | 1.0 | 1.2 | 23 | 5.1 | 2 | ✓ |
| 9 | 81 | 1.0 | 5.5 | 1.0 | 25 | 4.8 | 3 | ✓ |
| 10 | 81 | 1.0 | 1.5 | 4.0 | 25 | 4.8 | 1 | ✓ |
| 11 | 82 | 1.0 | 1.2 | 1.0 | 10 | 4.9 | 4 | ✕ |
| 12 | 100 | 1.1 | 1.0 | 1.0 | 10 | 10.0 | 5 | ✓ |

**Table 5.** Effect of the weights on the overall performance for a GPS-denied environment.

| Experiment No. | Weights | | | | | Performance | |
|---|---|---|---|---|---|---|---|
| | $r_c$ | $r_s$ | $r_a$ | $r_{ct}$ | $r_{oa}$ | #Collisions | Confined |
| 1 | 10 | 1.0 | 1.0 | 1 | 1 | 3 | ✕ |
| 2 | 100 | 5.0 | 1.5 | 5 | 1 | 4 | ✓ |
| 3 | 300 | 8.0 | 1.0 | 1 | 1 | 4 | ✓ |
| 4 | 500 | 6.0 | 1.0 | 1 | 1 | 5 | ✓ |
| 5 | 800 | 6.5 | 1.2 | 1 | 1 | 2 | ✓ |
| 6 | 750 | 5.5 | 1.0 | 1 | 1 | 1 | ✓ |
| 7 | 820 | 6.1 | 1.0 | 1 | 1 | 3 | ✕ |
| 8 | 800 | 6.2 | 1.5 | 1 | 1 | 1 | ✓ |
| **9** | **750** | **6.0** | **1.1** | **1** | **1** | **0** | **✓** |
| 10 | 800 | 6.0 | 0.9 | 1 | 1 | 1 | ✓ |
| 11 | 810 | 6.2 | 1.2 | 1 | 5 | 0 | ✕ |
| 12 | 800 | 6.2 | 1.0 | 3 | 1 | 1 | ✕ |

❶ Experiments 5 and 9 (highlighted in bold) show the base set of values of the weights for the GPS-aided and GPS-denied environments, respectively. We set these values as the default for the subsequent experiments.

❷     For the best case, the rule contributing the most to the final velocity is *cohesion*. Even though the values of $r_m$, $r_{ct}$, and $r_{oa}$ are much lower than $r_c$, the overall performance is quite sensitive to these values—this is also observed in Section 4.4.1.

### 4.3. Performance Analysis

To evaluate the performance of the proposed model, *SmrtSwarm*, in terms of the achieved flocking behavior, we ran the model in the simulated environment shown in Figure 10 with GPS enabled as well as disabled. We used a 10-drone swarm to begin with. As mentioned in Section 3.1.3, for a GPS-aided environment, we define a spherical boundary (radius = 30 m in the *x*, *y*, and *z* directions of Unity's coordinate system) around the leader drone as the confinement zone, whereas for GPS-denied environments, the field of view (FoV) of the drone becomes the confinement area. In our experiments, we used two cameras on all the drones, each with a field of view of 60°. Hence, the total FoV was 120° (similar to [43]). The swarm size, the simulation environment, the total FoV, and the confinement zone were the same for every experiment unless stated otherwise.

### 4.3.1. Swarming in a GPS-Aided Environment

We use two types of tags in *SmrtSwarm*: *Leader* and *Follower*. All the follower drones are given the *Follower* tag and the leader is given the *Leader* tag. The communication between drones is simulated using Unity's built-in shared variables. We uploaded a video of our simulations, which can be accessed using this link [44].

### 4.3.2. Swarming in a GPS-Denied Environment

In the real world, stereo cameras can directly compute the depth values of each pixel in the FoV. However, in Unity, the depth values (from simulated cameras) are stored in a z-buffer called a *depth buffer*. This buffer is stored in the GPU memory and is not directly accessible. We wrote a shader program using High-Level Shader Language (HLSL) to read the depth values [45]. The shader program gives the depth map as a $256 \times 256$ 2D matrix. They lie in the range of 0 to 1. We needed to post-process the data to transform them to match the camera's coordinate system. Furthermore, we also considered the camera's viewing range, which is 40 in the *x*, *y*, and *z* directions, and converted all normalized depth values to actual distances (in meters). In Figure 11, a few depth map illustrations are displayed. We make the following observations from the depth maps:

❶     The pixels within an object have similar depth values.

❷     We observe that clusters corresponding to obstacles are much larger than those of drones and have at least 2000 pixels. This defines a threshold for us—we use this to designate a cluster as an *obstacle*. Furthermore, obstacles, being static objects, often start from the bottom of the FoV.

❸     Also, there are a few clusters that correspond to random noise (far-away objects), which can be discarded if the total number of pixels forming a cluster is fewer than 8.

❹     As clear from Figure 11, some of the objects in the depth map may be occluded. Due to the fact that all the drones follow the flocking principles, there must be some distance between them and, as a result, a significant difference will be present in their depth values. This allows us to readily filter out each cluster even in the presence of occlusion.

We tried to design a proof technique for proving that our flocking rules will always maintain a coherent swarm and avoid collisions in all kinds of environments, regardless of obstacles. This is ongoing work and our results are not fully mature yet. We extensively searched the web, but we could not find any existing mathematical technique that similar papers have used. Research in drone swarming is validated using exhaustive experimentation, as we have performed [46–52].

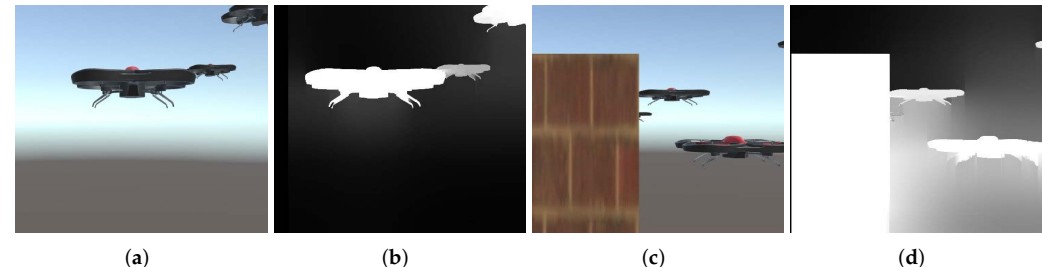

|     (a)     |     (b)     |     (c)     |     (d)     |

**Figure 11.** Depth maps of frames with and without obstacles. (**a**) Frame 1; (**b**) depth map of frame 1; (**c**) frame 2; (**d**) depth map of frame 2.

*4.4. Sensitivity Analysis*

To check whether the proposed model is robust enough, we ran the model in four different simulation environments (refer to Figure 12). These environments cover various lighting conditions, obstacle types, and relative positions of drones. The resulting swarm movement for all these cases is shown in an uploaded video [44]. We tuned the weights according to the scenes and list their final values in Tables 6 and 7. We make the following observations from the results:

❶    The weights are almost the same for all the environments.

❷    The model works well for almost all the environments if the value of the six-tuple $\langle r_c,$ $r_s, r_a, r_m, r_{ct}, r_{oa} \rangle = \langle 80, 1, 1, 1, 25, 5 \rangle$ for a GPS-aided environment.

❸    For a GPS-denied environment, the optimal value of the weight tuple is $\langle 750, 6, 1, 1, 1 \rangle$.

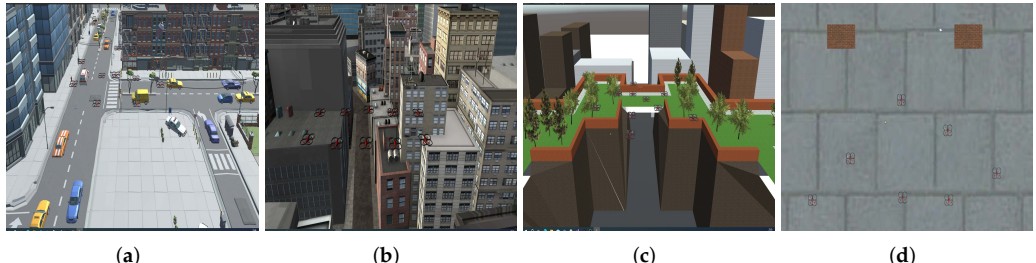

|     (a)     |     (b)     |     (c)     |     (d)     |

**Figure 12.** Simulation environments: (**a**) Scene 1; (**b**) scene 2; (**c**) scene 3; (**d**) scene 4.

**Table 6.** Weights for various simulation environments with GPS.

| Scene | Weights | | | | | | Performance | |
|---|---|---|---|---|---|---|---|---|
| | $r_c$ | $r_s$ | $r_a$ | $r_m$ | $r_{ct}$ | $r_{oa}$ | #Collisions | Confined |
| 1 | 78 | 1.0 | 1.0 | 1.0 | 21 | 5.0 | 0 | ✓ |
| 2 | 77 | 1.2 | 1.0 | 1.1 | 21 | 5.0 | 0 | ✓ |
| 3 | 77 | 1.1 | 1.5 | 1.0 | 25 | 5.0 | 0 | ✓ |
| 4 | 80 | 1.2 | 1.0 | 1.1 | 25 | 5.1 | 0 | ✓ |

**Table 7.** Weights for various simulation environments without GPS.

| Scene | Weights | | | | | Performance | |
|---|---|---|---|---|---|---|---|
| | $r_c$ | $r_s$ | $r_a$ | $r_{ct}$ | $r_{oa}$ | #Collisions | Confined |
| 1 | 750 | 6.1 | 1.0 | 1 | 1 | 0 | ✓ |
| 2 | 750 | 6.0 | 1.1 | 1 | 1 | 0 | ✓ |
| 3 | 700 | 6.2 | 1.5 | 1 | 1 | 0 | ✓ |
| 4 | 800 | 6.0 | 1.0 | 1 | 1 | 0 | ✓ |

4.4.1. Effect of the Proposed Rules

The proposed flocking rules in this paper are *migration*, *confinement*, and *obstacle avoidance*. To check whether these rules impact the overall swarming behavior, we ran

the model by disabling these rules individually in the simulation environment shown in Figure 12d (check the results in the uploaded videos here [44]). We make the following observations from the results:

❶ As per the migration rule, the drones migrated in the direction of the leader; after disabling this, the drones did not even move, and the significance of the migration force became abundantly clear.

❷ Without the obstacle avoidance force, drones collided with the obstacles.

❸ In the absence of the confinement force, all of the follower drones moved far ahead of the leader. However, when there was a confinement force, they remained confined within a boundary.

### 4.5. Scalability Analysis

To check the scalability of the proposed model, we varied the swarm size by keeping the simulation environment fixed. When the size of our swarm increased, we increased the radius ($\delta$) of the confined region around the leader so that the swarm could cover a larger area and we reduced the likelihood of a collision. However, in the case of GPS-denied environments, there is no concept of a confinement zone. We tuned the weights in this case as well, and we list the optimal values in Tables 8 and 9. We make the following observations from the results:

❶ The weights were almost the same for all swarm sizes.

❷ For the GPS-aided environment, the model worked well with all the swarm sizes if the weight values $\langle r_c, r_s, r_a, r_m, r_{ct}, r_{oa} \rangle = \langle 80, 1, 1, 1, 25, 5 \rangle$. The results are in line with the observations made in Section 4.4.

❸ Similarly, for the GPS-denied environment, the optimal values of weights are the same as given in Section 4.4.

**Table 8.** Weights for drone swarms of different sizes in a GPS-aided environment.

| Experiment No. | Swarm Size | Radius ($\delta$) | Weights | | | | | | Performance | |
|---|---|---|---|---|---|---|---|---|---|---|
| | | | $r_c$ | $r_s$ | $r_a$ | $r_m$ | $r_{ct}$ | $r_{oa}$ | #Collisions | Confined |
| 1 | 5 | 30 | 81.0 | 1.0 | 1.2 | 1.0 | 25 | 4.8 | 0 | ✓ |
| 2 | 7 | 35 | 79.0 | 1.1 | 1.5 | 1.1 | 25 | 4.7 | 0 | ✓ |
| 3 | 8 | 35 | 80.0 | 1.0 | 1.3 | 1.0 | 22 | 5.0 | 0 | ✓ |
| 4 | 10 | 40 | 79.5 | 1.0 | 1.4 | 1.1 | 24 | 4.8 | 0 | ✓ |
| 5 | 12 | 45 | 80.0 | 1.2 | 1.4 | 1.0 | 23 | 5.0 | 0 | ✓ |
| 6 | 15 | 50 | 79.0 | 1.1 | 1.4 | 1.0 | 24 | 5.0 | 0 | ✓ |

**Table 9.** Weights for drone swarms of different sizes in a GPS-denied environment.

| Experiment No. | Swarm Size | Weights | | | | | Performance | |
|---|---|---|---|---|---|---|---|---|
| | | $r_c$ | $r_s$ | $r_a$ | $r_{ct}$ | $r_{oa}$ | #Collisions | Confined |
| 1 | 5 | 790 | 6.0 | 1 | 1.0 | 1.0 | 0 | ✓ |
| 2 | 7 | 808 | 6.2 | 1.2 | 1.0 | 1.0 | 0 | ✓ |
| 3 | 8 | 810 | 6.0 | 0.9 | 1.0 | 1.0 | 0 | ✓ |
| 4 | 10 | 800 | 6.5 | 1.0 | 1.0 | 1.0 | 0 | ✓ |
| 5 | 12 | 795 | 6.0 | 1.0 | 1.0 | 1.0 | 0 | ✓ |
| 6 | 15 | 800 | 6.0 | 1.0 | 1.0 | 1.0 | 0 | ✓ |

### 4.6. Real-Time Performance of SmrtSwarm

To check the performance of the proposed model, *SmrtSwarm*, in a real-world environment, we ran it on a **Beaglebone Black board** [53]. Beaglebone Black is a popular embedded board with an ARM Cortex-A8 processor clocked at 1 GHz frequency. It also has 512 MB RAM. We used Python 3.8 and GCC version 4.9.2 to implement the swarming model. Table 10 shows the execution time of each step involved in the swarming model on the board. We make the following observations from the results:

❶ For a GPS-aided environment, all the steps have an extremely low latency ($<0.3$ ms). Additionally, the variance in execution times is very low ($<2\%$).

❷ The previously mentioned observation (point (1)) holds true in a GPS-denied environment as well, except for two steps: object detection and obstacle avoidance. The maximum and average latencies for these steps vary significantly across frames because these values are directly proportional to the number of objects in the depth map.

❸ The step that takes the longest (with a maximum value of ≈12 ms) is object detection in the depth map using our algorithm.

❹ The total latency for the GPS-aided environment is very low (<0.5 ms). The FPS (frames processed per second) can be as high as 2000 frames per second, which is orders of magnitude more than what is required (we typically need 10–20 FPS; 75 FPS is considered to be high given that traditional displays operate at 30 FPS) for drones, which are relatively slow-moving. Even for a GPS-denied environment, the maximum frame rate that can be achieved is 75 FPS (total execution time < 14 ms).

**Table 10.** Runtime (in milliseconds) breakdown of our proposed method.

| Steps | Environment | | | | | |
| | GPS-Aided | | | GPS-Denied | | |
| | Max | Min | Avg | Max | Min | Avg |
| --- | --- | --- | --- | --- | --- | --- |
| Object detection | - | - | - | 11.87 | 8.17 | 9.95 |
| Cohesion | 0.04 | 0.01 | 0.02 | 0.02 | 0.01 | 0.01 |
| Separation | 0.28 | 0.21 | 0.24 | 0.27 | 0.23 | 0.25 |
| Alignment | 0.01 | 0.01 | 0.01 | 0.02 | 0.01 | 0.01 |
| Migration | 0.01 | 0.01 | 0.01 | - | - | - |
| Confinement | 0.05 | 0.03 | 0.04 | 0.01 | 0.01 | 0.01 |
| Obstacle avoidance | 0.02 | 0.02 | 0.02 | 1.23 | 0.84 | 1.03 |
| Total time | 0.44 | 0.39 | 0.42 | 13.55 | 09.32 | 11.44 |

## 5. Conclusions

In this work, we proposed a leader–follower flocking model for controlling a drone swarm, aiming to enhance coordination within the swarm. To achieve this, we introduced three additional rules—migration, confinement, and obstacle avoidance—to the traditional Reynolds flocking model. These rules play a crucial role in maintaining better coordination and synchrony among the drones.

While GPS-assisted communication is effective for calculating the target velocity of each drone under ideal conditions, we recognize the limitations posed by unreliable GPS signals in real-world scenarios. To address this challenge, we presented a depth-map-based approach that allows for accurate control and coordination of nearby drones even in the absence of reliable GPS signals. This alternative approach significantly enhanced the swarm's operational capabilities, enabling precise coordination and control in various environments.

In addition to our model's contributions to swarm coordination and overcoming GPS limitations, it is essential to consider the evaluation of countermeasures and defensive strategies against adversarial actions. By studying the interactions between the swarm and moving adversaries, valuable insights can be gained into adversarial tactics, strategies, and vulnerabilities. These insights can further guide the development of more robust defense mechanisms and contribute to the creation of resilient swarm behaviors. This is a part of future work.

**Author Contributions:** N.B.: Conceptualization, methodology, software, writing—original draft. H.V.: Conceptualization, methodology, software, writing—original draft. A.D.: Conceptualization, methodology, validation, writing—review and editing, formal analysis. B.B.: Conceptualization, methodology, supervision. S.R.S.: Conceptualization, methodology, validation, formal analysis, writing—review and editing, supervision. All authors have read and agreed to the published version of the manuscript.

**Funding:** This work was funded jointly by the governments of India and New Zealand (No. MI02544G).

**Data Availability Statement:** The source code and all simulation results are available on a GitHub repository [54].

**Conflicts of Interest:** The authors declare no conflict of interest.

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
