# Peer review of "SmrtSwarm: A Novel Swarming Model for Real-World Environments"

_drones, doi:10.3390/drones7090573_

Round 1

Reviewer 1 Report

Comments and Suggestions for Authors

I found this paper very interesting. You introduced the problem clearly and invoked interest for the reader in your introduction.   The abstract can be improved to set the context for readers who are not familiar with drone swarms.  I suggest that you reword the abstract to provide more context and motivation for the problem before mentioning Reynold's model. 

On page 3, I suggest that you explain Figure 1 process with more clarity to establish that 'the process' is generic and referring to previous flocking models.  In Figure 1, should members be neighbours? The steps in Figure 1 could be explained in more detail. 

Please define CNNs (page 6)

Page 7, Migration Rule - "In these models" - does that refer to leader-follower-based models? perhaps you might say 'In such models' or 'In similar models' to be clearer.  

I was a little confused when reading about whether all drones share a goal.  In your introduction you mention that drone swarming can be defined as coordinated behaviour of autonomous drones .. achieve a common goal. I had presumed that drones had some awareness of this common goal? But perhaps not, because on page 7 only the leader is aware of the intended goal (optimal route to an implied destination)?

In section 3.1.3 you refer to 'described earlier' I think it might assist the reader if you name the earlier section( 3.1.2/previous section)?

Would it be possible to move Figure 4 to page 10?

Page 10. Define reference drone.

Page 11. you mention static obstacles very briefly without clear explanation. Do you assume that all moving objects are drones in the swarm and there are now competing drones or moving obstacles? Perhaps you can clarify this.

Please explain what you mean by 'implicitly follow it or have their own guidance system' page 11.  Do you mean the leader drone has the goal to follow the deer or an implicit goal to follow a visual target?

Section 3.2.3.1 and 3.2.3.2 I suggest you include 'for GPS denied environment' in the name of these sections. 

Section 3.2.3.2 page 13. Please explain 'original movement' is that the most recent movement ? Over what time period?

Section 4.4 I wonder if you can comment qualitatively about what the weights in the optimal weight tuple suggest ?

Comments on the Quality of English Language

Overall, the paper is well written and easy to read. Please review the expression/language used in section 3.2.3.1 on page 12 to improve the explanation of the alignment rule. 

Reviewer 2 Report

Comments and Suggestions for Authors

This paper presents a captivating exploration of enhancing coordination within drone swarms through the introduction of a leader-follower flocking model enriched with migration, confinement, and obstacle avoidance rules. Your innovative approach holds great potential for reshaping the landscape of swarm behavior and control.

This paper effectively conveys the significance of the challenge which are  addressed and outlines the methodological framework you've developed.  The idea of employing a depth map-based approach to circumvent the limitations of GPS-assisted communication in real-world scenarios is both novel and commendable. This approach introduces a fresh perspective to the field, which undoubtedly holds promise for overcoming practical challenges faced by swarm control systems.

While the paper has successfully elucidated the central concepts and  methodology, it will be particularly interesting in gaining a deeper understanding of the practical implementation and performance of your proposed model. In light of this, I would like to kindly inquire whether the authors might consider sharing the simulation results in video format and associated materials in a GitHub repository. The value of such an endeavor lies in its potential to provide fellow researchers and enthusiasts with access to the outcomes of your work, enabling a more comprehensive assessment and engagement with your contributions.

Reviewer 3 Report

Comments and Suggestions for Authors

The manuscript  propose a new set of rules and a game-theoretic method to set the values of the hyperparameters to design robust swarming algorithms for a complex environments

In the abstract ,authors give the problem ,the solution ,and implement their algorithms on the Unity gaming engine and a Beaglebone Black board .Readers can easily understand the innovative points of the article.

In Section 1 and 2  authors introduced the background  and analyzed a large amount of related works and provided their own contributions.

In Section 3 ,authors illustrate the details  of the proposed swarming model.

In Section 4 ,extensive experiments show that the effectiveness and superiority of the proposed algorithms .

The manuscript is well written and  organized. However, I have the following concerns.

1,According to  template of  drones  ,tables in the paper should be formatted using three horizontal lines.The title of the table should be placed above the table. 

2.,According to  template of  drones  ,authors should provide some information about Author Contributions,Funding,Data Availability Statement,and etc .

3.According to the template of drones, the title of a figure with multiple panels is not correct.

4.In the Section 3.2.2,there is a sentence ,After the completion of the steps mentioned in Sections 3.2.1 and 3.2.2,.... However, the specific steps are not clearly stated in the paper.

Round 2

Reviewer 2 Report

Comments and Suggestions for Authors

Paper is revised well and it is ready for publishing.